# OpenReview forum: "Understanding Jailbreak Success: A Study of Latent Space Dynamics in Large Language Models"
_ICLR.cc/2025/Conference — Submitted to ICLR 2025_

### Official Review · Reviewer_5pMp · 2024-10-27

**Soundness:** 3
**Presentation:** 4
**Contribution:** 3
**Rating:** 6
**Confidence:** 3

**Summary:**

This paper provides an empirical understanding about jailbreak success, specifically focused on the activation vectors. Authors hypothesized that jailbreak attacks with different attacks will have lead to similar underlying intermediate vectors from the language model, and this "jailbreak vector" can be used to promote or suppress the harmful behaviors. Their empirical results validate their claim, showing that (1) different jailbreak steering vectors have high cosine similarities (in Figure 3) and (2) these vectors can be transferred so that jailbreak vector from tactic A can be used to make model safer on tactic B (and other tactics).

The paper further discussed about the hypothesis that jailbreaking attacks lead LMs fail to detect the harmfulness of the prompt, which is previously argued in Zou et al, 2023. The paper showed the result of PCA anaylsis (in Figure 5) and cosine similarity plots (in Figure 6) to show that jailbreaking often leads to the reducing the harmful features from the harmful instructions. But it also mentioned that low harmfulness always lead to low ASR score.

**Strengths:**

The paper is well-written and the claim is clear.

This paper presents clear empirical observations through multiple experiments about their claim that jailbreak vector exists. The authors tested on multiple LMs (7B-14B models) and tested multiple jailbreak tactics. Also, this claim is meaningful to understand the jailbreak success, and can be used to steer LMs to behave safer without additional fine-tuning.

The paper seems not to over-claim but tried to present their results objectively. For example, in Section 5.4, authors discussed about the harmfulness suppression, mentioning the exceptional cases where low harmfulness scores do not lead to low ASR scores. This helps readers and future researchers to understand the phenomenon in better ways.

**Weaknesses:**

I can't find significant weaknesses from the paper, though I think the paper could be improved if the authors include automatic jailbreak attacks like PAIR or TAP. These attacks don't specify the type of attacks but LLMs automatically find the attack. I think understanding their jailbreak success using latent space dynamics can be helpful to build successful defenses on such attacks.

**Questions:**

- There are bunch of automatic jailbreaking attacks that are trying not to use pre-defined tactics but discover different tactics using LLMs or other optimization-based approaches (for example, TAP or PAIR.) Do these attacks conclude to similar results?
- I wonder this observation can be also applicable to the safety-trained models -- for example, Llama2 or Llama3 are known to have low ASR on well-known jailbreak attacks due to safety training such as SFT or RLHF. Have you tried studying the vectors from these models?

---

> ### Author Response · Authors · 2024-11-22
>
> Thank you very much for the thoughtful and comprehensive review! In the following we would like to address the two questions you had.
>
> **On comparison with automatic jailbreak attacks:**\
> We agree that it is interesting to study automatic jailbreak attacks. This is why we included the automatic jailbreak attack GCG in our analysis, which finds a prompt-specific suffix that leads the model to jailbreak. TAP and PAIR also optimize the prompt to get the answer to a specific harmful question. The difference to GCG is that the prompts based on TAP and PAIR are human-understandable. Given our inclusion of the GCG jailbreak attack in our analyses, we expect our results to transfer to other automatic jailbreak attacks like PAIR and TAP.
>
> **On studying safety-tuned models:**\
> The models we include in our analyses do have effective safety alignment of different forms, while still being jailbreakable. We agree that it will be interesting to study strongly aligned models, once there is a meaningful number of efficient jailbreaks for these newer models. Please refer to the global comment for a detailed explanation of our model choice.

---

> > ### Comment · Reviewer_5pMp · 2024-11-26
> >
> > Thank you for your response. I will keep my rating as is.

---

### Official Review · Reviewer_2nR9 · 2024-11-03

**Soundness:** 3
**Presentation:** 2
**Contribution:** 2
**Rating:** 5
**Confidence:** 3

**Summary:**

This paper aims to understand existing jailbreak methods over LLMs by analyzing the latent space dynamics. First, they constructed jailbreak vectors based on the activations of LLMs calculated on harmful datasets and harmless datasets. They find that the jailbreak vector from a single class of jailbreaks can also work to mitigate jailbreak effectiveness from other semantically-dissimilar classes. Second, they analyze the evolution of cosine similarity between harmfulness direction and activations at each token position for one harmful question without jailbreak (none) and for different jailbreak types. They find that successful jailbreaks have significantly lower representations of harmfulness at the end of instruction for most models, which indicates that the jailbreaks suppress the harmfulness feature on the prompts.

**Strengths:**

1.	It is interesting to use jailbreak vectors for analysis. The paper also demonstrated that jailbreak vectors can  effectively prevent the success of jailbreaks across different types via activation steering, pointing to a shared underlying mechanism.

2.	The paper discussed the concept of models’ perception of prompt harmfulness, and suggested that some jailbreaks succeed by reducing the perception of prompt harmfulness, preventing the refusal response.

**Weaknesses:**

1.	The presentation of methodology part can be improved by providing a workflow figure. This can help people better understand how and where the jailbreak vectors are calculated, and also how the mitigation is performed.

2.	The findings in Section 5.2 need to be re-evaluated carefully. They find that jailbreak steering vectors have a positive cosine similarity with one another, and hypothesize that jailbreak vectors from one class will work to steer away from successful jailbreaks of other classes. However, the positive cosine similarity could result from the representation degeneration issues, which find that the representations learned by Transformer models tend to clustered in a cone in representation space.

3.	Subtracting jailbreak vectors may reduce ASR, but also potentially destroy the model parameters, making it hard for practice use. For example, in line 371, it may induce content repetition in the response. In addition, it also better if the performance of a general task can be included in this part for comparison.

**Questions:**

1.	In model settings, are the included LLMs trained with RLHF for safety issues? Will this affect the validity of analysis in this paper?

2.	The paper said no sampling is used when decoding. However, in practice, LLM based chatbots usually use sampling for diversity. This discrepancy may affect the conclusions in this paper. A study on the temperature selection can also verify the robustness of the proposed method.

3.	The paper uses middle layers for calculating jailbreak. Did you try the lower or upper layers? Or maybe an ablation study can further enhance our understanding.

4.	In Section 4.4, the harmless dataset was generated by instructing ChatGPT. Were these samples checked by human annotators to guarantee the data quality?

5.	In line 256, the LLMs do not behave the same. What affects this, data or architecture?

---

> ### Author Response · Authors · 2024-11-22
>
> Thank you for the comprehensive feedback! We hope our arguments and additional analyses will fully address the weaknesses and questions.
>
> **On the workflow figure:** New workflow figure is included on page 4.
>
> **On representation degeneration:**\
> Thanks for pointing out that a positive cosine similarity between jailbreak steering vectors may result from representations clustering in a cone. We carefully re-evaluated Section 5.2 by looking at cosine similarity values of our steering vectors with i) an “italian” steering vector and ii) a vector steering the model to more “happy” behavior (data from [1]). The italian vector is more closely related to our jailbreak vector setup, for which we contrasted harmful questions in English and Italian. For the happiness vector we contrast the last token activations of happy and sad statements, averaged over 203 pairs. Both the italian and happiness vectors show noticeably reduced similarity to our jailbreak vectors, which stresses the meaningfulness of their similarity. We added the results and their discussion to our paper (pages 6 and 20).
>
> **On evaluating steering performance on unrelated tasks:**\
> It is true that subtracting the jailbreak vectors leads to a decrease in ASR scores with some reduction in answer quality in the form of repetitions (but not for all answers). Similarly, subtracting our normalized vectors during inference on harmless questions reduces performance (slightly) on the MMLU benchmark (Vicuna 7B: -1.4%points, Vicuna 13B: 0.3%p., Qwen1.5 14B Chat: -16.2 %p., MPT 7B Chat: -2.8%p.). Searching for the optimal steering parameters like a dynamic intervention strength and layer can help to reduce quality issues (see e.g. [2]). We did not conduct such a thorough analysis since the goal of our paper is to study shared underlying jailbreak mechanisms and not to find the optimal defense strategy (see global comment). We included the benchmark results in our paper, clarified the paper’s main focus, and highlighted areas where further research on defense methods is needed (page 8). For future research we would be interested to see work that extracts a shared underlying component of our jailbreak vectors, which could then be used in a steering defense framework.
>
> **On model safety training:**\
> We chose differently aligned models for our analysis, including models aligned with fine-tuning, DPO and PPO. Our analyses indicate the transferability of results across differently aligned models.
>
> **On testing different temperature and layer values:**\
> While the focus of our paper is on studying jailbreak mechanisms, we agree that it is valuable to think how our jailbreak vectors can be employed for defense (see global comment). Repeating our analysis for Qwen1.5 14B Chat with a temperature of 0.7 leads to very similar results compared to when no sampling is used (0.7 is the default value in commonly deployed LLM chatbots, see e.g. [3]).
> Looking at the layers 4, 15, 25 and 40, we observe that steering at layer 4 is by orders of magnitudes less successful than steering at the middle layer, while steering at layer 40 mainly leads to non-understandable outputs. In contrast, using steering vectors from different middle layers (15 and 25) results in comparable steering outcomes as with the middle layer, albeit with slightly less strong mitigation successes. These additional results corroborate our analyses and are in accordance with the steering literature, finding the middle layers to be most successful for steering [4,5,6] (as mentioned in our paper). We added these additional explorations to our paper (pages 8, 22). However, we would like to reiterate that finding the optimal defense strategy based on these steering vectors is not within the scope and interest of our paper and requires more research.
>
> **On checking quality for harmless dataset:**\
> Yes, the responses of ChatGPT to generate the harmless dataset were checked by the authors to ensure quality.
>
> **Differences in model behavior:**\
> Since we analyze a different number of jailbreak types for the models, clustering differences in our PCA analysis could be influenced by both data and architecture. Some clusters could also be more separable along a third PCA dimension. We added this consideration to the paper (page 5). However, we want to point out that the PCA clustering is only a preliminary step in our analysis and that the main part of the paper reveals consistent behavior across models.
>
> [1] Zou, A., ... & Hendrycks, D. (2023). Representation engineering: A top-down approach to AI transparency. \
> [2] Marshall, T., ... & Belrose, N. (2024). Refusal in LLMs is an Affine Function.\
> [3] https://lmarena.ai \
> [4] Arditi, A., ... & Nanda, N.. Refusal in language models is mediated by a single direction. \
> [5] Panickssery, N., ... & Turner, A.. Steering Llama 2 via contrastive activation addition. \
> [6] Turner, A., ... & MacDiarmid, M. (2023). Activation addition: Steering language models without optimization.

---

> > ### Comment · Reviewer_2nR9 · 2024-12-01
> >
> > Thanks for your response. I will raise the rating.

---

> > > ### Author Response · Authors · 2024-12-02
> > >
> > > Great that we could address your concerns. Is there anything else that we can clarify?

---

### Official Review · Reviewer_c6to · 2024-11-03

**Soundness:** 2
**Presentation:** 2
**Contribution:** 2
**Rating:** 3
**Confidence:** 4

**Summary:**

This paper investigates how different types of jailbreaks in LLMs might share underlying mechanisms, making it possible to counter multiple jailbreaks using shared “jailbreak vectors.” The study finds that effective jailbreaks tend to reduce a model’s perception of harmfulness, suggesting a way to develop more resilient safeguards against these manipulations.

**Strengths:**

Strengths
1.	Some new Insights on Jailbreaks: the paper explores how different jailbreak types in LLMs might share common mechanics, providing a relatively fresh view on understanding and countering them.
2.	Possible Mitigation Solution: By identifying shared jailbreak properties, the study suggests ways to create more robust defenses against jailbreak attacks.

**Weaknesses:**

Weaknesses
1.	Limited Model Range: The study focuses on only a few models, which may make its conclusions less generalizable. Moreover, the choice of Vicuna lacks sufficient justification, as it may not adequately represent the current SOTA among 7B/13B models.
2.	Related work is not highly related to the research topic of this paper and did not include the most recent studies on understanding the how jailbreak works in the LLMs.

**Questions:**

The authors said "layer 16 for 7B and layer 20 for 13B and 14B parameter models" in line 162, my question is "Multiple layers, rather than a single layer, should be considered as the middle layers. what if we select a different middle layer, e.g., layer 14 for 7B, layer 24 for 13B. will this have some influences on the results?"

---

> ### Author Response · Authors · 2024-11-22
>
> Thank you for your review and for acknowledging that we provide a fresh view on understanding jailbreak success mechanisms. We are addressing the weaknesses and questions next.
>
> **On including Vicuna and model selection:** \
> We agree that it is always better to include more models in experiments. However, our choice of models covers a range of models with different alignment strategies and sizes, which increases the generalizability of our results. We include Vicuna in our analysis as it has been shown to be susceptible to jailbreaks, while refraining to answer harmful requests that do not use jailbreak techniques. This makes Vicuna an interesting model to study, and it can furthermore function as a comparison to our other models that are aligned via RLHF and DPO (for further comments on model selection see global comment).
>
> **On citing literature:**\
> The literature we cite is highly related to our research as we mention papers that we directly build on and that hypothesize how jailbreaks work. Could you please indicate which papers you find missing?
>
> **On selection of middle layer:**\
> The main goal of our jailbreak steering analysis is to understand whether different jailbreaks work via a similar mechanism. Our results point to such a shared underlying mechanism for different model classes using the middle layer of each model. We focus on the middle layer because research shows that these are the layers where models process high level concepts and where steering is effective [1, 2, 3]. Our finding that we can successfully prevent jailbreaks via steering is a by-product of our work and gives first indications for a successful jailbreak prevention strategy via jailbreak steering vectors. We agree that the development of an optimal defense strategy needs more research, which includes finding the most effective intervention layer and strength of intervention. However, finding these optimal parameters is a separate research endeavor, which is distinct from investigating the existence of shared underlying jailbreak mechanisms.
> To answer the posted question, we repeat our analyses with layers 4, 15, 25, and 40 for the Qwen1.5 14B Chat model. The results show that steering is comparably effective (albeit less strong) for the other middle layers 15 and 25, significantly less effective for layer 4, and not at all effective for layer 40 as the output is greatly distorted (repetition of single tokens for many examples). Hence, we are able to replicate that steering is most effective at middle layers. We added these supporting results to the paper (pages 8, 22) and made it more explicit in the text, that we are not directly proposing a jailbreak defense method (see global comment).
>
> [1] Andy Arditi, Oscar Obeso, Aaquib Syed, Daniel Paleka, Nina Panickssery, Wes Gurnee, and Neel Nanda. Refusal in language models is mediated by a single direction. arXiv preprint arXiv:2406.11717, 2024.
>
> [2] Nina Panickssery, Nick Gabrieli, Julian Schulz, Meg Tong, Evan Hubinger, and Alexander Matt Turner. Steering Llama 2 via contrastive activation addition. arXiv preprint arXiv:2312.06681, 2023.
>
> [3] Turner, A. M., Thiergart, L., Leech, G., Udell, D., Vazquez, J. J., Mini, U., & MacDiarmid, M. (2023). Activation addition: Steering language models without optimization. arXiv e-prints, arXiv-2308

---

> > ### Comment · Reviewer_c6to · 2024-11-26
> >
> > Thanks for the response. However, I also share the concern that subtracting the jailbreak vector may not be a good idea, as it can lead to unexpected side effects, with utility drops varying across different LLMs (1 or 2 pt might be acceptable, more than 5 pt is not acceptable). Hence, I decide to keep my score.

---

> > > ### Author Response · Authors · 2024-11-27
> > >
> > > To clarify, we are using vector subtraction as a tool to investigate mechanism overlap/shared features across jailbreaks, and are not advocating for using this as a inference-time anti-jailbreak intervention as-is.

---

### Official Review · Reviewer_yyNu · 2024-11-09

**Soundness:** 3
**Presentation:** 3
**Contribution:** 3
**Rating:** 5
**Confidence:** 3

**Summary:**

This paper explores how jailbreaking techniques work by analyzing latent space dynamics in large language models. The authors found that different types of jailbreaks might share a common internal mechanism and that a jailbreak vector from one class can mitigate others, and that effective jailbreaks can reduce the model's perception of prompt harmfulness. These findings provide valuable insights for understanding jailbreaks and developing better safeguards in large language models.

**Strengths:**

* This paper provides a new perspective to understand jailbreak by analyzing latent space dynamics in large language models. Their findings are very interesting that different types of jailbreaks might share a common internal mechanism and a jailbreak vector can be used to improve/reduce safety of large language models.
* The paper covers many different kinds of jailbreak methods, making it more convincing that there may exist a common internal mechanism in large language jailbreaking.

**Weaknesses:**

* This paper only evaluate 4 models and some of them are not well aligned, e.g. Vicuna is only trained on ShareGPT dataset without RLHF/DPO Stage. In addition, these four models do not behave consistently, e.g. Vicuna shows better cluster patterns than Qwen 14B Chat. In addition, from Vicuna 7B to 13B, I also find that 13B model activation cluster is scattered. This made me question that when the model goes to large sizes and more aligned, will these patterns be less obvious and visible? Hence, I think the authors may compare more aligned models like Llama 3.1 in various sizes and study if they follow similar patterns.

* The authors study how to use the  jailbreak vector to reduce/improve jailbreak success. My question is that in practice, how do we use such techniques? If we minus jailbreak vectors for both harmful and harmless questions, will it reduce the models' helpfulness/correctness/truthfulness? It will be interesting to know these results in general benchmarks like MMLU or AlpacaEval.

**Questions:**

See comments in Weaknesses.

---

> ### Author Response · Authors · 2024-11-22
>
> Thank you for the thoughtful and comprehensive review and for acknowledging that our paper provides a new perspective and interesting findings to better understand jailbreak success mechanisms. We hope that our comments and edits to the paper help to fully address the weaknesses mentioned.
>
> **On the safety alignment of our model selection:** \
> It is correct that the Vicuna models are not aligned via an additional RLHF step. However, the models refuse to answer the 352 harmful questions of our dataset to 98% (Vicuna 13B v1.5) and 90% (Vicuna 7B v1.5), showing that there is effective alignment in place. In order to further explore whether differently aligned models show similar behavior, we additionally analyze Qwen1.5 14B Chat, which is aligned via DPO and PPO [1] as well as MPT 7B Chat, which is aligned using the HH-RLHF dataset by Anthropic [2, 3]. Hence, we cover models with different alignment strategies, which we view as a strength of our paper. As mentioned in the global comment, we did not include more aligned models like the Llama 3 models as they have been trained to refuse the suit of jailbreaks we are considering, given the fast moving nature of the field. In order to construct a variety of meaningful jailbreak vectors that we can compare across different model families, we deliberately decided to focus on more jailbreakable but aligned models. We are convinced that the consistency of our results across these differently aligned models is helpful to increase the general understanding of jailbreak mechanisms (see reasoning in global comment).
>
> **On the behavior of different model sizes:**\
> We agree that there are slight differences between the models in how pronounced the
> clustering patterns are in the PCA analysis, which we point out in the paper. Next to architectural differences, deviating clustering patterns can also result from the different number of jailbreak types that are considered for the different models. Furthermore, some clusters might be more separable along a third PCA dimension. We added these considerations to the paper (page 5). However, we would like to emphasize that the PCA clustering serves as an exploratory step in the analysis and is not the central focus of this work. The core findings of the paper highlight consistent patterns across the models, demonstrating the transferability of jailbreak steering vectors across different classes. This suggests the presence of a shared underlying mechanism. Additionally, the steering vector analyses reveal stronger patterns for bigger models: Qwen1.5 14B Chat uses SOTA alignment techniques and performs best in these analyses. Hence, we expect that if one could construct jailbreak vectors from powerful jailbreaks that break strongly aligned models, the steering results would be equally or even more pronounced.
>
> **On using jailbreak steering vectors as a defense method:**\
> We thank the reviewer for seeing the potential of using our jailbreak steering vectors as a defense method in practice. As outlined in the global response, we agree that more research needs to be conducted to find optimal steering parameters like the intervention strength. This is important because simply subtracting our normalized jailbreak vectors from harmless questions in the MMLU benchmark can lead to more or less pronounced performance decreases (Vicuna 7B: -1.4%points, Vicuna 13B: 0.3%points, Qwen1.5 14B Chat: -16.2 %points, MPT 7B Chat: -2.8%points). However, more sophisticated steering frameworks exist, which tailor the intervention strength to the amount of a feature already present and account for a constant bias in the representation, resulting in less distortion on benign inputs (see e.g. [4]). We leave the development of such a defense framework to future research since the main goal of our paper is to study shared underlying jailbreak mechanisms. We included the benchmark results in our paper, clarified the paper’s main focus, and highlighted areas where further research on defense methods is needed (page 8).
>
> [1] Qwen Team. (2024, February). Introducing Qwen1.5. https://qwenlm.github.io/blog/qwen1.5/
>
> [2] https://huggingface.co/datasets/Anthropic/hh-rlhf
>
> [3] https://www.databricks.com/blog/mpt-7b
>
> [4] Marshall, T., Scherlis, A., & Belrose, N. (2024). Refusal in LLMs is an Affine Function. arXiv preprint arXiv:2411.09003.

---

> > ### Comment · Reviewer_yyNu · 2024-11-26
> >
> > Thank you for your response. I will keep my score as is.

---

### Author Response · Authors · 2024-11-22
**Global response to all reviewers**

We are delighted that the reviewers recognize our paper as providing an important contribution to understanding jailbreak success mechanisms, which we believe to be a critical step toward developing more robust jailbreak mitigation techniques.

We view our main contributions as the following:
- Successfully constructing jailbreak steering vectors that work to mitigate the success of other jailbreak types, providing evidence for a shared underlying mechanism.
- Demonstrating that this underlying mechanism could be the jailbreaks’ ability to reduce the models’ perception of prompt harmfulness at the end of instruction.
- Investigating a comprehensive set of different jailbreak types and models that are aligned via different alignment strategies.

In the reviewers’ comments, we identified two main themes that we would like to address globally, in addition to providing detailed responses to individual reviewers. These themes are:
1. Justifying the model choice.
2. Using jailbreak vectors as a defense strategy.

**Concerning point 1: model choice justification**\
Some reviewers were unsure whether the models we chose underwent safety training and were curious whether our results transfer to more strongly aligned models like Llama 3. Regarding the safety training question, the four models we selected are aligned via finetuning, RLHF, or DPO, thereby representing a range of different alignment strategies. To ensure these models were adequately aligned, we checked their refusal rates on 352 harmful scenarios without a jailbreak, showing that the models are capable of refusing the vast majority of these harmful scenarios. If a model was not capable of refusing a specific harmful request, we excluded this request from the steering analyses.

Our criteria for model selection was such that we included models of different sizes that are aligned but still susceptible to jailbreaks. This choice is determined by the primary goal of our study: to investigate the mechanisms by which jailbreaks succeed. Our selection criteria also explains why we did not include strongly aligned models like Llama 3 since they are robust to our diverse set of jailbreaks. Hence, studying the newest, strongly aligned models would limit our ability to analyze a meaningful range of effective jailbreak types.

We argue that our focus on susceptible-yet-aligned models is relevant given the lack of general understanding concerning how jailbreaks work. Furthermore, one can expect that models that appear robust today may become more jailbreakable in the future. By providing evidence for a shared underlying mechanism across different jailbreak types and model alignment strategies, our findings provide insights that are likely to generalize to other aligned models.

**Concerning point 2: using jailbreak vectors as a defense strategy** \
We were pleased to see that some reviewers recognized the potential of using our steering vectors as a jailbreak defense strategy. We agree that evaluating the effectiveness of such an approach in practice is crucial, particularly in terms of maintaining response quality on harmless queries. It is furthermore necessary to find the optimal intervention layer and strength as pointed out by some reviewers. However, the primary focus of our paper is on studying the underlying dynamics of jailbreaks, rather than on the comprehensive development or assessment of a defense strategy. We view our work as a foundational step that can inform future research on jailbreak-vector-based defenses. We made this distinction clearer in our paper.

To answer the reviewers’ questions on whether our results transfer to different steering settings, we repeated our analyses for different layers, a temperature value different from zero, and evaluated a possible steering implementation on the MMLU benchmark. The results of these analyses are discussed in detail in our individual responses to the reviewers. While we incorporated these findings into the paper, as they strengthen our findings, we would like to reiterate that the main contribution of our work is in increasing our understanding of jailbreak dynamics across various jailbreak types and models. This focus does not allow for an exhaustive development of an optimal jailbreak defense strategy, which is an important but distinct line of future research.

We would like to thank the reviewers again for their valuable feedback and think that we are able to address their remaining concerns in the individual responses and the changes we made to the paper.

---

### Meta-Review · Area_Chair_mg3q · 2024-12-21

**Metareview:**

This paper studying common underlying mechanisms for different types of jailbreaks in language models and how they can be potentially used to counteract jailbreaking. The reviewers agreed that the paper makes an interesting contribution to our understanding of jailbreak success mechanisms. Also, they agree that finding that jailbreak steering vectors can be used to mitigate the success of other jailbreak types is interesting and has potential implications for developing jailbreak defenses.

Weakness: The reviewers have primarily cited two reasons to reject -- (1) limited number of models used in the study (which that authors have justified, and reviewers didn't raise more concerns), and (2) the defense strategy could have unintended consequences like reduced general capabilities. The authors have argued that understanding jailbreaking is the main problem they study, and defense is not the objective of this paper. Despite this claim, defense is central to main story of this paper as it currently stands and does not seem supplementary, it is mentioned as early as page 1 (figure 1). Shifting focus will require major rewrite and hence, I recommend rejection.

The authors should consider the following suggestions for revision:

1. Include a broader range of models in their study, including models from different alignment strategies and sizes.
2. Provide a more thorough justification for their choice of models.
3. Expand the related work section to include the most recent studies on understanding how jailbreaks work in LLMs.
4. Conduct a more comprehensive analysis of the potential negative effects of using jailbreak vectors as a defense strategy.

**Additional Comments On Reviewer Discussion:**

See meta review

---

### Decision · Program_Chairs · 2025-01-22

Reject